# Respiratory and intestinal epithelial cells exhibit differential susceptibility and innate immune responses to contemporary EV-D68 isolates

**Megan Culler Freeman[1,2†], Alexandra I Wells[1,2†], Jessica Ciomperlik-Patton[3], Michael M Myerburg[4], Liheng Yang[1,2], Jennifer Konopka-Anstadt[3], Carolyn B Coyne[1,2]\***

[1]Department of Pediatrics, Division of Infectious Diseases, UPMC Children's Hospital of Pittsburgh, Pittsburgh, United States; [2]Center for Microbial Pathogenesis, UPMC Children's Hospital of Pittsburgh, Pittsburgh, United States; [3]Division of Viral Diseases, Centers for Disease Control and Prevention, Atlanta, United States; [4]Pulmonary, Allergy and Critical Care Medicine, University of Pittsburgh School of Medicine, Pittsburgh, United States

**Abstract** Enterovirus D68 (EV-D68) has been implicated in outbreaks of severe respiratory illness and is associated with acute flaccid myelitis (AFM). EV-D68 is often detected in patient respiratory samples but has also been detected in stool and wastewater, suggesting the potential for both respiratory and enteric routes of transmission. Here, we used a panel of EV-D68 isolates, including a historical pre-2014 isolate and multiple contemporary isolates from AFM outbreak years, to define the dynamics of viral replication and the host response to infection in primary human airway cells and stem cell-derived enteroids. We show that some recent EV-D68 isolates have decreased sensitivity to acid and temperature compared with earlier isolates and that the respiratory, but not intestinal, epithelium induces a robust type III interferon response that restricts infection. Our findings define the differential responses of the respiratory and intestinal epithelium to contemporary EV-D68 isolates and suggest that a subset of isolates have the potential to target both the human airway and gastrointestinal tracts.

**\*For correspondence:**
carolyn.coyne@duke.edu

**Present address:** †Department of Molecular Genetics and Microbiology, Duke University School of Medicine, Durham, United States

**Competing interests:** The authors declare that no competing interests exist.

## Introduction

Enteroviruses (EVs) are a family of positive-stranded RNA viruses, including coxsackieviruses, echoviruses, enterovirus A71 (EV-A71), and enterovirus D68 (EV-D68) that are responsible for a broad spectrum of illness in humans. EVs, specifically EV-D68 and EV-A71, have been associated with acute flaccid myelitis (AFM), a polio-like illness causing paralysis in previously healthy individuals, primarily children, which has peaked in even numbered years from at least 2014 until 2018 (*Messacar et al., 2015*; *Midgley et al., 2015*; *Mishra et al., 2019*; *Schubert et al., 2019*). While 2020 was anticipated to be a peak year for AFM, there was not a surge of cases reported, indicating that coronavirus infection-control measures such as social distancing and mask usage also diminished exposure to other circulating pathogens (*CDC, 2020*). While EVs are traditionally spread via the fecal-oral route, but previous work with EV-D68 isolates before the AFM outbreak in 2014 suggested reduced replication in acidic environments and improved replication at lower temperature than traditional EVs, suggesting suitability for respiratory tract replication (*Oberste et al., 2004*).

EV-D68 has undergone rapid evolution since the 1990s, leading to the emergence of four clades, termed A–D (*Du et al., 2015*; *Tokarz et al., 2012*). This degree of evolution has led to loss of

neutralization from pre-existing antibodies, highlighting the potential significance of these changes (*Imamura et al., 2014*). Contemporary EV-D68 isolates exhibit different biological properties than historical reference isolates, including replication in neuronal cells (*Brown et al., 2018*). EV-D68 is often detected in patient respiratory samples; however, EV-D68 genetic material has also been detected in stool specimens and wastewater, suggesting presence in the digestive tract and potential transmission by the fecal–oral route (*Bisseux et al., 2018*; *Pham et al., 2017*; *Weil et al., 2017*). The viral and host determinants that influence EV-D68 tropism remain largely unknown, particularly in the respiratory and gastrointestinal (GI) epithelium. Moreover, whether there are differences in the replication dynamics and/or host responses to isolates circulating prior to AFM outbreaks versus contemporary isolates is also unclear.

The EV-D68 reference isolate Fermon is often used as a historic isolate, due to its isolation in the mid-1960s. However, this isolate has undergone decades of passage through cell lines and has thus likely undergone changes that make it well-adapted for replication in cell culture and less representative of its original sequence when it was isolated from a child with pneumonia (*Schieble et al., 1967*). These changes highlight the need to perform comparative studies using pre-outbreak and contemporary EV-D68 isolates in order to define the viral and host determinants of infection. In this study, we performed comparative studies of replication kinetics, temperature sensitivity, polarity of infection, and cellular responses to infection using a panel of EV-D68 isolates, including a historic isolate and multiple isolates from AFM outbreak years. To define host cell-type-specific differences in EV-D68 replication and/or host responses, we performed comparative studies in primary human bronchial epithelial (HBE) cells grown at an air–liquid interface and in primary human stem-cell-derived intestinal enteroids. We found that respiratory and intestinal cell lines were permissive to both historic and contemporary EV-D68 isolates, but that there were isolate-specific differences in temperature sensitivity at 33°C or 37°C. In contrast, primary HBE cells were largely resistant to EV-D68 replication, with only one isolate, KY/14/18953, able to replicate. KY/14/18953 and MA/18/23089 were able to replicate in human enteroids. Primary HBE, but not enteroids, mount a robust innate immune response to EV-D68 infection, characterized by the induction of type III interferons (IFNs) and, to a lesser extent, type I IFNs. Lastly, we show that inhibition of IFN signaling enhances EV-D68 replication in primary HBE, supporting a role for this signaling in the control of viral replication in the airway. Collectively, these data define the differential responses of the respiratory and intestinal epithelium to historic and contemporary EV-D68 isolates.

## Results

### EV-D68 replication in lung and intestinal cell lines varies with isolate and temperature

We sought to evaluate the replication competency of a panel of EV-D68 isolates, including a historical 2009 isolate and five contemporary isolates from outbreaks in the AFM peak years of 2014 and 2018 in cell lines representing the respiratory and intestinal tracts (details of viral isolates can be found in *Supplementary file 2*). To do this, we used the MD/09/23229 isolate, collected in 2009 and in clade A, as a reference isolate prior to the 2014 outbreak and multiple isolates associated with AFM outbreak seasons, including 2014 and 2018. These isolates are inclusive of multiple clades, B1, B2, B3, and D (*Du et al., 2015*; *Hadfield et al., 2018*; *Sagulenko et al., 2018*; *Sun et al., 2019*), that have been associated with peak-year AFM outbreaks. In addition, KY/14/18953 and US/IL/18952 isolates are paralytogenic in mouse models (*Brown et al., 2018*; *Hixon et al., 2017*). A phylogenetic comparison of isolates used in this study is shown in *Figure 1A*. Sequence comparisons between these isolates revealed that consistent with previous work (*Tokarz et al., 2012*; *Wang et al., 2019*), the greatest sequence divergence between isolates exists in the VP1 structural protein, with additional divergence in the VP2 and VP3 regions (*Figure 1B*, *Supplementary file 1* ).

We evaluated replication at 33°C and 37°C in Calu-3 cells, a lung adenocarcinoma cell line, and in Caco-2 cells, a colon adenocarcinoma cell line. We found that while all isolates replicated to some degree at 33°C in Calu-3 cells, two isolates, MD/09/23229 and MA/18/23089, were unable to efficiently replicate at 37°C (*Figure 2A*). In Caco-2 cells, all isolates were able to replicate at 33°C (*Figure 2B*). When infections were performed at 37°C, MA/18/23089 and KY/14/18953 continued to replicate well over background in Caco-2 cells (*Figure 2B*). In contrast, infections in Calu-3 cells

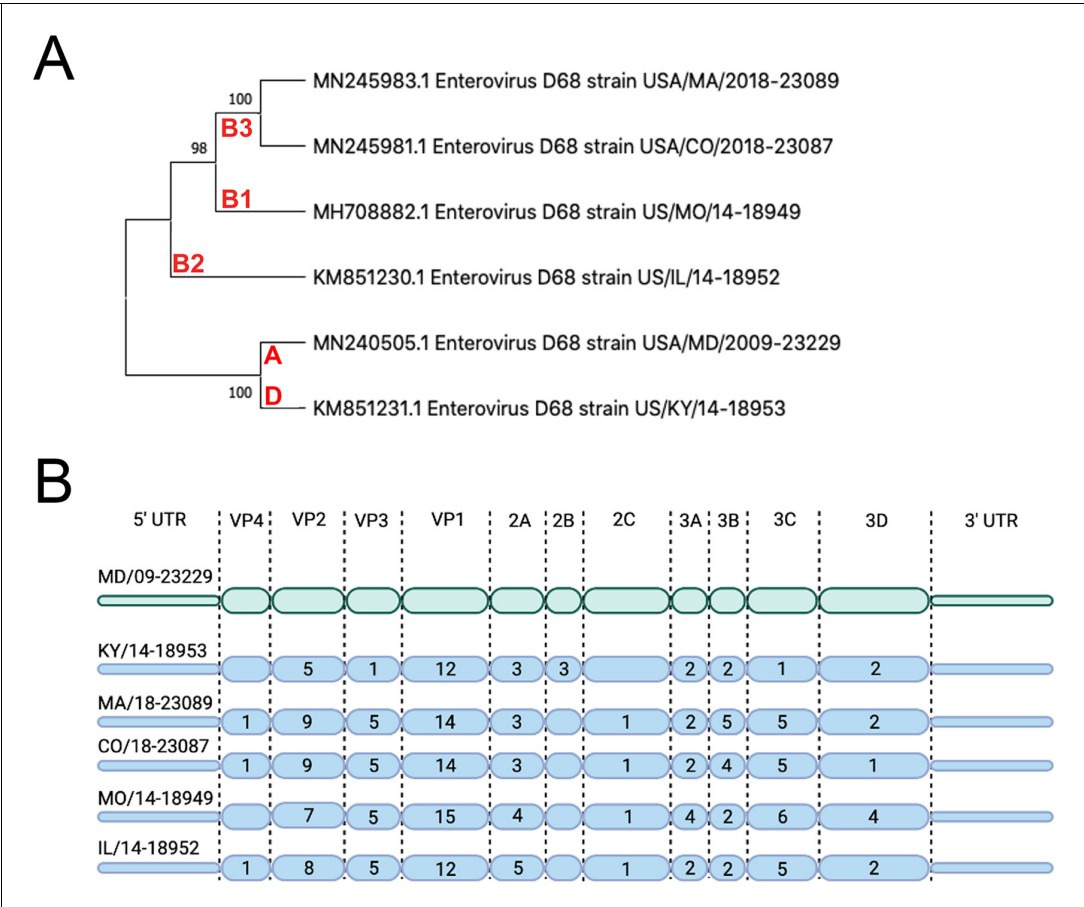

**Figure 1.** Evolutionary relationship of EV-D68 isolates. (**A**) Phylogenetic tree of EV-D68 isolates used in this study. USA/MD/2009–23229 in clade A represents the pre-AFM isolate, while the isolates from clades B1, B2, B3, and D were circulating in peak AFM years. Optimal tree is shown and drawn to scale, with branch lengths in the same units (base substitutions per site) as the evolutionary distances used to construct the tree. Percentage next to the branches is replicate trees (1000 replicates) in which associated taxa clustered together via bootstrap test. Clade denoted in red. (**B**) Polyprotein diagram of EV-D68, with MD/09–23229 reference isolate represented on top in green. Isolates associated with AFM years are listed below in blue. Numbers within the protein schematic represent the number of amino acid mutations within that protein as compared to reference isolate.

performed at 37°C severely restricted the replication of several isolates, including MD/09/23229, IL/14/18952, and MO/14/18949, whereas the replication of KY/14/18953 was less severely restricted (*Figure 2A*). Collectively, these studies suggest that select EV-D68 isolates exhibit cell-type-specific sensitivity to temperature in cell lines (summarized in *Figure 2C*).

## Some contemporary EV-D68 isolates have increased acid tolerance

We found that all EV-D68 isolates were capable of replicating in GI-derived cell lines. However, in addition to cellular tropism, enteric viruses must be stable in acidic environments to infect the GI tract. Previous work suggested that select EV-D68 isolates were destabilized following exposure to low pH (pH 4–6) as a mechanism of genome release during vial entry (*Liu et al., 2018*). However, whether this instability might influence the enteric route of transmission is unclear. In order to define the stability of EV-D68 virions in various conditions that mimic the environment in the GI tract, we exposed historical and contemporary isolates of EV-D68 to simulated intestinal fluids of the stomach and fed and fasted states of the small intestine over short (30 min) and long (60–120 min) exposure times. These fluids reflect not only the differential pH of the GI tract, but also contain bile acid and phospholipids that better recapitulate some aspects of the GI luminal content. To compare the stability of EV-D68 to other members of the enterovirus family that are transmitted primarily via the fecal–oral route, we performed similar studies with echovirus 11 (E11) and EVA71. E11 and EVA71 were stable in both fed state small intestine (FeSSIF pH 5) and fasted state small intestine (FaSSIF

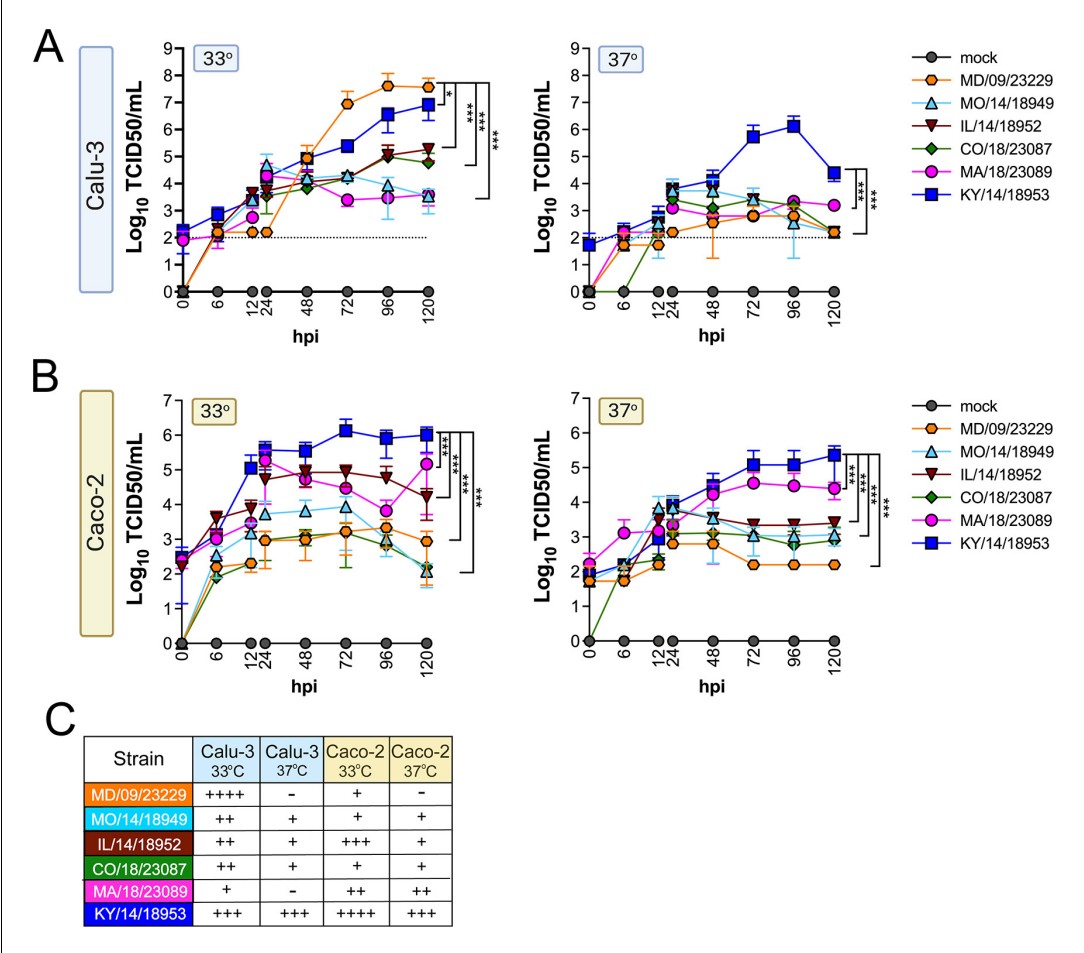

**Figure 2.** EV-D68 replication in lung and intestinal cell lines varies with strain and temperature. Calu-3 cells (**A**) or Caco-2 cells (**B**) were infected with the EV-D68 strains MD/09/23229 (orange), MO/14/18949 (cyan), IL/14/18952 (burgundy), CO/18/23087 (green), MA/18/23089 (pink), or KY/14/18953 (blue) at an MOI of 5 and incubated at 33°C or 37°C. The supernatant was sampled at the indicated hours post-infection (hpi) and titrated by TCID50. Data are shown as mean ± standard deviation from three replicates. Dotted line denotes limit of assay detection. (**C**) Summary table denotes titers at 72 hpi, + corresponds to $10^3$, ++ $10^4$, +++ $10^5$, and ++++ $10^6$. Significance was determined by a two-way ANOVA with multiple comparisons. *p<0.05, **p<0.005, ***p<0.0005, ****p<0.0001 compared to the KY/14/18953, which exhibited the highest replication levels.

pH 6.5) for all exposure times tested (*Figure 3A,B*). However, whereas E11 exhibited significant reductions in titer when exposed to fasted state simulated gastric fluid (FaSSGF pH 2.0), EV71 was less impacted by this exposure (*Figure 3A,B*). None of the EV-D68 isolates tested were able to withstand the most acidic fluid (FaSSGF, pH 2.0) (*Figure 3C–F*). However, whereas EV-D68 isolates were generally stable in FaSSIF pH 6.5 conditions (*Figure 3C–F*), there were isolate-specific differences in stability in FeSSIF pH 5 conditions, with KY/14/18953 and to a lesser extent MA/18/23089 exhibiting some stability in this condition (*Figure 3D–F*). These data suggest that some contemporary isolates of EV-D68 exhibit enhanced stability in low pH conditions.

## Comparison of EV-D68 growth characteristics in primary human airway epithelial cells and stem-cell-derived enteroids

We found that many EV-D68 isolates efficiently infected airway- and intestinal-derived cell lines, which occurred in a temperature-dependent manner in some cases (*Figure 2*). However, given that cell lines do not fully recapitulate the complexities of the airway and intestinal epithelium, we performed similar studies in primary cell models. To model the human airway, we used primary HBE cells grown at an air–liquid interface (ALI). HBE have increased similarity to the human respiratory tract with respect to polarization, functional cilia, and mucus production than cell-line-derived

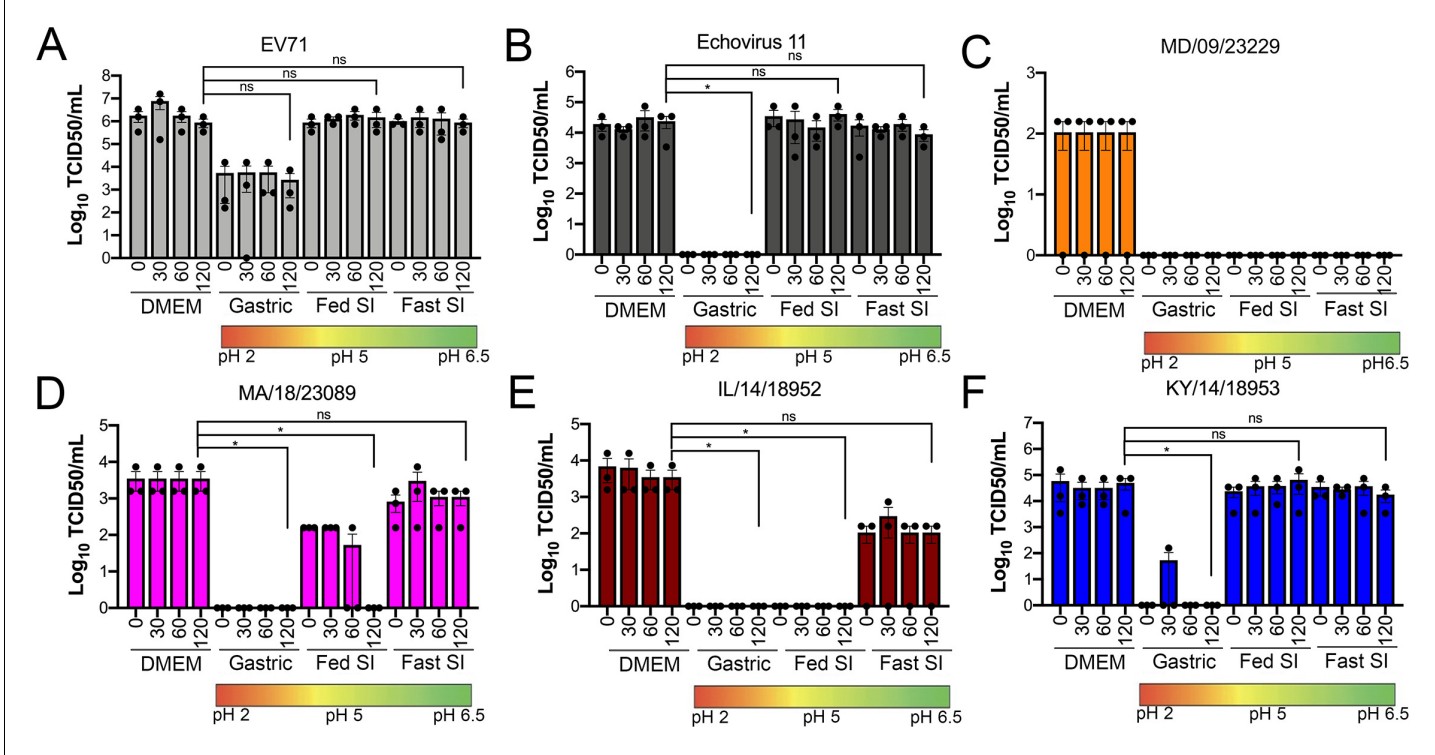

**Figure 3.** Select contemporary EV-D68 strains exhibit increased acid tolerance. (**A**) Enterovirus 71 (EV71), (**B**) echovirus 11, or (**C–F**) the indicated EV-D68 isolates ($10^6$ PFU/ml) were incubated with control medium (DMEM), pH 2 FaSSGF (Gastric), pH 5 FeSSIF (Fed SI), or pH 6.5 FaSSIF (Fast SI) solution and incubated at 37°C for the indicated times. An aliquot of the virus/fluid mixture was collected, neutralized with sodium hydroxide, and then evaluated for infectivity via TCID50 assays. Titers are shown as mean ± standard deviation from three independent replicates. Significance was assessed at 120 min post-incubation using a one-way ANOVA compared to DMEM-incubated controls, *p<0.05, ns, not significant.

respiratory models and thus provide a more physiological system to study EV-D68 infections in the human airway. We infected HBE cells from at least two independent donors from either the apical or basolateral domains and first measured viral titers to determine whether EV-D68 exhibited a polarity of entry. First, we measured titers in the apical supernatants and found that one isolate, KY/14/18953, infected similarly from the apical and basolateral domains at 33°C or 37°C, but exhibited preferential release from the apical surface (*Figure 4A–E*). In contrast, MD/09/23229 replicated more efficiently from the basolateral surface and exhibited a temperature preference for 33°C (*Figure 4A–E*). Another contemporary isolate, IL/14/18952, infected best from the basolateral surface at 33°C while the contemporary isolate MA/18/23089 infected primary HBE inefficiently from either domain or temperature (*Figure 4A–E*). Next, we measured titers from the basolateral supernatants to determine if EV-D68 exhibited preferential egress from the apical or basolateral cell surface. We found that all isolates that replicated efficiently in primary HBE exhibited low to undetectable titers in basolateral supernatants (*Figure 5A–E*) compared to those detected in apical supernatants (*Figure 4*). These data suggest that some isolates exhibit a preferential polarity of infection and are released primarily via the apical surface.

Next, we determined whether EV-D68 could infect GI-derived primary cells, particularly given that all isolates replicated to high titers in a GI-derived adenocarcinoma cell line (Caco-2). To do this, we used human primary stem cell-derived enteroids, which we previously used to define the cellular tropism of other enteroviruses in the GI epithelium (*Drummond et al., 2017*; *Good et al., 2019*). We found that only one isolate, KY/14/18953, replicated in human enteroids, which occurred in a temperature-independent manner whereas there were very low levels of infection by other isolates tested, although MD/09/232229 exhibited some capacity to replicate to low levels (*Figure 6A–B*). A limitation of the above-described model is that enteroids grown in Matrigel exhibit an 'inside

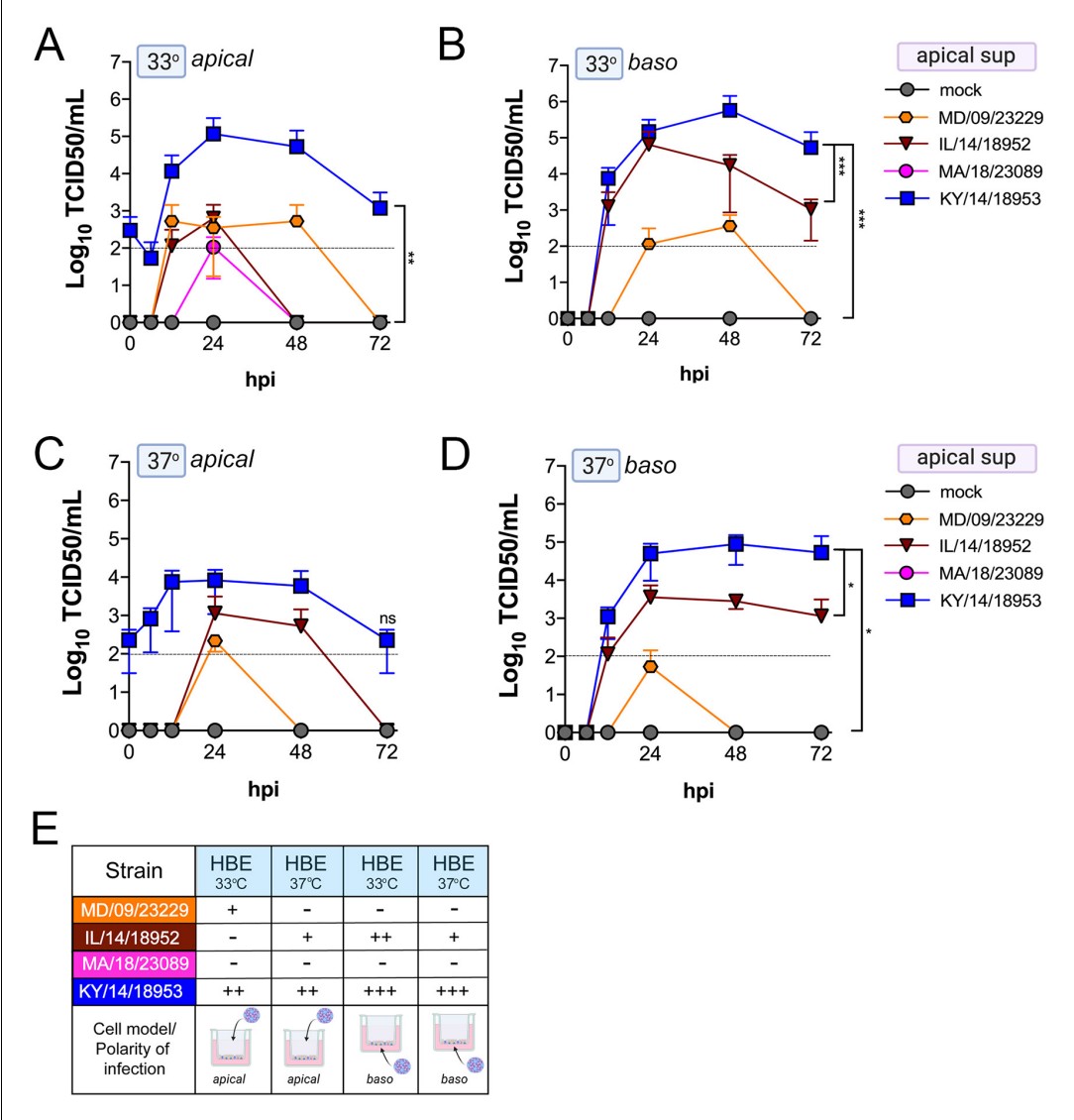

**Figure 4.** Comparison of EV-D68 strain-specific growth characteristics in apical supernatants of primary human airway epithelial cells. Primary human bronchial epithelial (HBE) cells grown at an air–liquid interface were infected with $10^6$ PFU of the indicated EV-D68 strains: MD/09/23229 (orange), IL/14/18952 (burgundy), MA/18/23089 (pink), or KY/14/18953 (blue) and incubated at 33˚C (**A, B**) or 37˚C (**C, D**) for the indicated hours post-infection (hpi). HBE were infected from either the apical (**A, C**) or basolateral (**B, D**) surfaces. Supernatants were sampled at the indicated hpi from the apical compartment and titers determined by TCID50 assays. Titers are shown as mean ± standard deviation from three independent replicates. Dotted line denotes limit of assay detection. (**E**) Summary table denotes titer at 48 hpi as collected from the apical compartment, - indicates no detectable replication, + corresponds to $10^3$, ++ $10^4$, +++ $10^5$, and ++++ $10^6$. Significance determined by two-way ANOVA. *$p<0.05$, **$p<0.005$, ***$p<0.0005$, compared to the KY/14/18953, which exhibited the highest replication levels.

out' polarity, with the luminal surface facing inward. As we have previously shown that some entero-viruses such as EV-A71 exhibit preferential infection of the apical domain, we next determined whether EV-D68 exhibited a similar polarity, which might explain the low levels of infection in enter-oids grown in Matrigel (*Good et al., 2019*). To address this, we cultured intestinal crypts on Trans-well inserts, which allows for the development of a monolayer containing diverse intestinal cell types (*Good et al., 2019*). Similar to our studies in HBE, we infected intestinal monolayers from the apical (*Figure 6C,E*) or basolateral (*Figure 6D,F*) domains and sampled the apical (*Figure 6C,D*) or baso-lateral (*Figure 6E,F*) supernatant for infectious virus. We focused on KY/14/18953 and MA/18/23089 for these studies because they replicated the best in Caco-2 cells at 37˚C compared to other isolates.

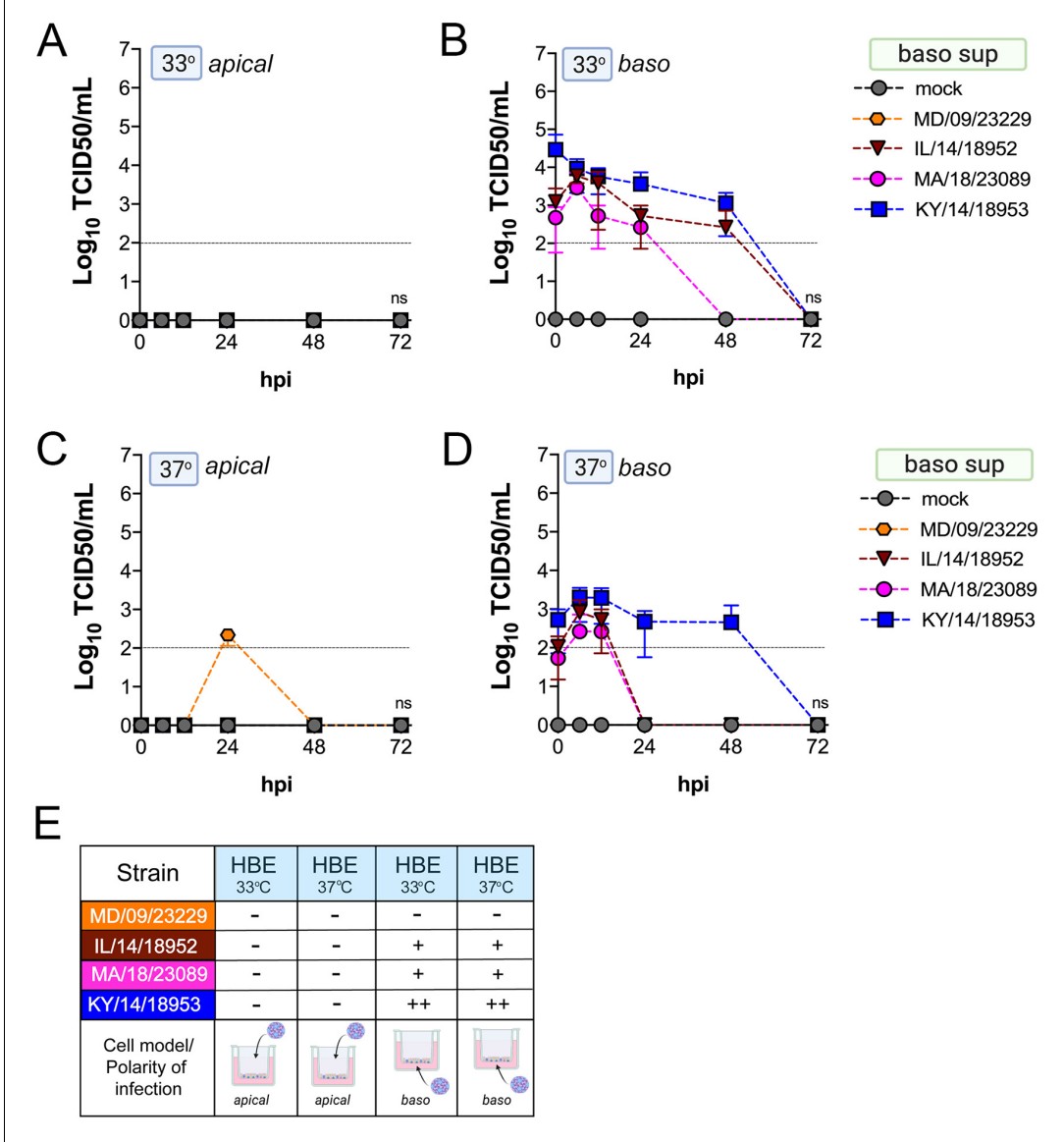

**Figure 5.** Comparison of EV-D68 strain-specific growth characteristics in basolateral supernatants of primary human airway epithelial cells. Primary human bronchial epithelial (HBE) cells grown at an air-liquid interface were infected with $10^6$ PFU of the indicated EV-D68 strains: MD/09/23229 (orange), IL/14/18952 (burgundy), MA/18/23089 (pink), or KY/14/18953 (blue) and incubated at 33°C (A, B) or 37°C (C, D) for the indicated hours post-infection (hpi). HBE were infected from either the apical (A, C) or basolateral (B, D) surfaces. Supernatants were sampled at the indicated hpi from the basolateral compartment and titers determined by TCID50 assays. Titers are shown as mean ± standard deviation from three independent replicates. Dotted line denotes limit of assay detection. (E) Summary table denotes titer at 48 hpi as collected from the basolateral compartment, − indicates no detectable replication, + corresponds to $10^3$, ++ $10^4$, +++ $10^5$, and ++++ $10^6$. Significance determined by two-way ANOVA compared to the KY/14/18953, which exhibited the highest replication levels, ns not significant.

We found that KY/14/18953 replicated to high titers when inoculated from either the apical or basolateral surfaces but exhibited a preferential release into the apical compartment (*Figure 6C–F*), similar to what was observed in primary HBE. In contrast to our findings in Matrigel-derived enteroids, we found that MA/18/23089 replicated to high titers when infection was initiated from the apical surface, with slightly lower titers from the basolateral domain (*Figure 6C–F*). However, similar to KY/14/18953, this isolate also exhibited preferential release into the apical compartment (*Figure 6C–F*). Collectively, these data show that some contemporary isolates of EV-D68, particularly KY/14/18953, can replicate to higher titers in both primary HBE and enteroids. In contrast, the historical isolate

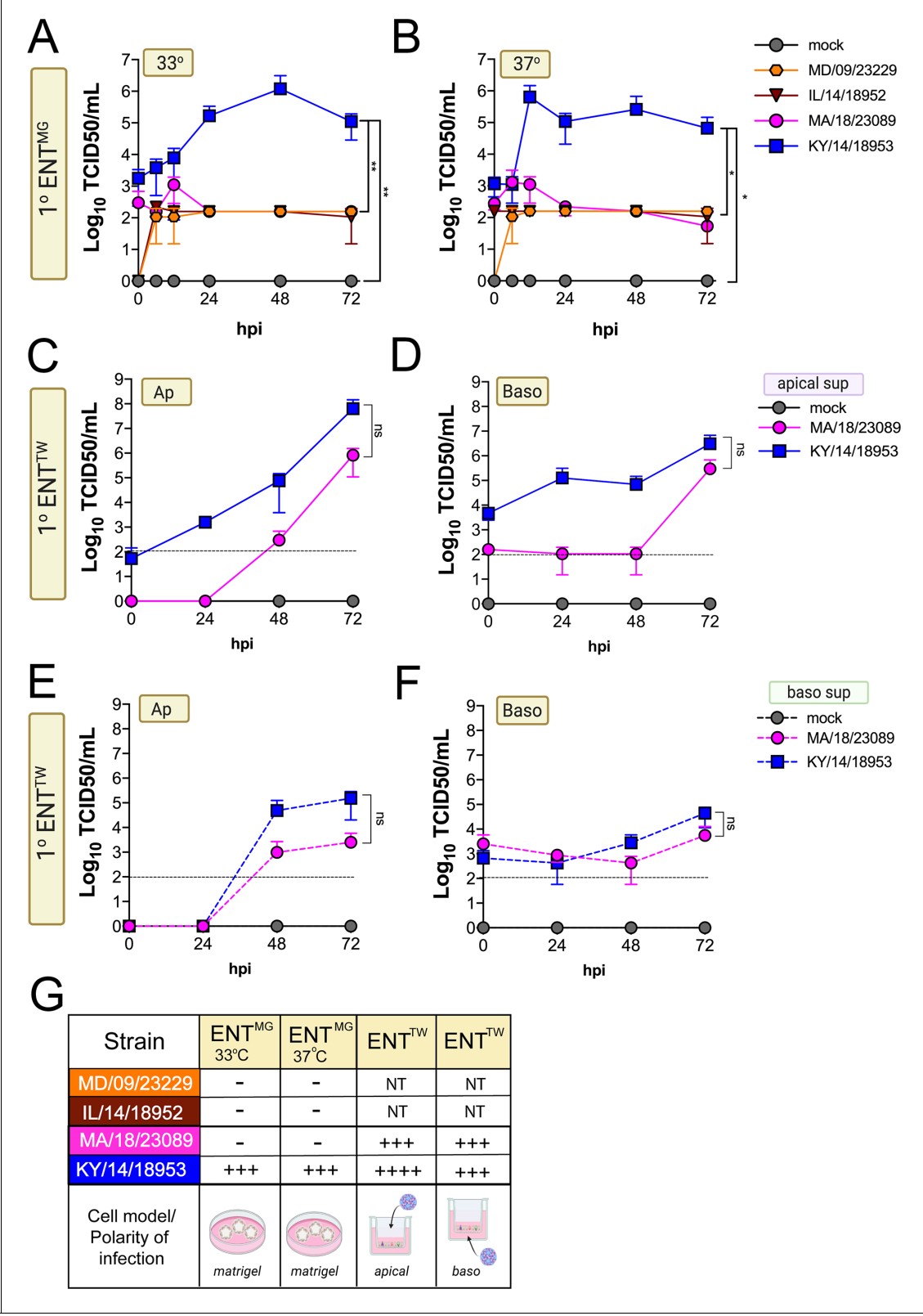

**Figure 6.** Comparison of EV-D68 strain-specific growth characteristics in primary human enteroids. Primary enteroids grown on Matrigel (**A, B**) or transwells (**C–F**) were infected with $10^6$ PFU of the indicated EV-D68 strains: MD/09/23229 (orange), IL/14/18952 (burgundy), MA/18/23089 (pink), or KY/14/18953 (blue) and incubated at 33°C (**A**) or 37°C (**B–F**) for the indicated hours post-infection (hpi). For intestinal cells grown on transwells, cells were infected from either the apical (**C, E**) or basolateral (**D, F**) surfaces. Supernatants were sampled at the indicated hpi from both apical (**C, D**) or

*Figure 6 continued on next page*

*Figure 6 continued*

basolateral (**E, F**) compartments and titers determined by TCID50 assays. Titers are shown as mean ± standard deviation from three independent replicates. Dotted line denotes limit of assay detection. (**G**), Summary table denotes titer at 72 hpi, + corresponds to $10^3$, ++ $10^4$, +++ $10^5$, and ++++ $10^6$, NT not tested. Significance determined by one-way ANOVA. *$p < 0.05$, **$p < 0.005$, ns, not significant.

MD/09/23229 replicated to low titers in primary HBE, which only occurred at 33°C and was unable to replicate in enteroids.

## EV-D68 infection induces cell-type-specific antiviral signaling

To define the cellular response to EV-D68 infection in HBE and in enteroids, we first performed RNA-Seq-based whole transcriptional profiling using select EV-D68 isolates, the historic strain MD/09/23229 and, due to successful replication under all tested conditions, KY/14/18953. Consistent with our infectious titer data, HBE cells infected from the basolateral surface had higher viral RNA (vRNA) fragments per kilobase per million reads mapped (FPKM) reads than those infected apically (*Figure 7A*). However, despite near-equivalent viral input, HBE cells infected with MD/09/23229 had higher vRNA FPKM values than those infected with KY/14/18953 (*Figure 7A*), even though cells infected with KY/14/18953 had higher infectious titers. We found that vRNA FPKM values in enteroids infected with KY/14/18953 were significantly higher than those observed in HBE and that these values were independent of temperature, as we obtained similar values in enteroids infected at 33°C or 37°C (*Figure 7A*). Next, we performed differential expression analysis to identify transcripts induced by EV-D68 infection. Despite significant differences in the levels of infection, HBE infected with either MD/09/23229 or KY/14/18953 from the basolateral surface induced similar numbers of transcripts, with MD/09/23229 inducing 178 (*Figure 7B*, *Supplementary file 4*) and KY/14/18953 inducing 189 (*Figure 7C*, *Supplementary file 4*). Consistent with the low levels of vRNA present in HBE infected from the apical surface, relatively very few transcripts were induced under these conditions, with MD/09/23229 inducing 37 (*Figure 7B*, *Supplementary file 4*) and KY/14/18953 inducing 30 (*Figure 7C*, *Supplementary file 1B*). Of the transcripts induced by basolateral infection (*Figure 7D*), approximately half were shared between HBE infected with MD/09/23229 or KY/14/18953 (92 total, *Supplementary file 5*). These transcripts were enriched in interferon-stimulated genes (ISGs) (*Figure 7F*, *Supplementary file 5*). In contrast, there were very few transcripts induced by both HBE and enteroids infected with KY/14/18953, with only nine transcripts shared between these conditions, despite enteroids inducing a greater total number of transcripts (332 total) (*Figure 7E*, *Supplementary file 6*). Of these transcripts, five included ISGs (MX2, IFIT3, IFIT1, IFI27, and IFITM1), which were induced in all conditions tested (*Figure 7G*). Consistent with the induction of ISGs, HBE infected with MD/09/23229 or KY/14/18953 from the basolateral surface, and to a lesser extent the apical surface, potently induced the expression of the type III IFNs IFN-λ1–3, but not type I or II IFNs (*Figure 7H*). In contrast, KY/14/18953 infection of enteroids elicited no significant induction of these transcripts, despite the higher levels of vRNA present in these samples (*Figure 7A,H*). Taken together, these data suggest that there are cell-type-specific differences in the response of HBE and enteroids to EV-D68 infection.

## EV-D68 infection of primary human airway cells preferentially induces type III IFNs

Our RNASeq-based studies pointed to cell-type-specific differences in the transcriptional response of primary HBE and enteroids to EV-D68 infection. To further define the cellular response to EV-D68 infection, we performed multianalyte Luminex-based assays for 37 pro-inflammatory cytokines in cells infected with historical and contemporary EV-D68 isolates at 33°C at both 24 hr and 48 hr post-infection. Importantly, this approach allowed for confirmation that the cytokines induced at the transcript level were also secreted at the protein level. In HBE, we also directly compared the impact of the polarity of infection on cytokine induction. At 24 hr p.i., all EV-D68 isolates induced the type III IFNs, IFN-λ1 and IFN-λ2, with little to no significant induction of type I IFNs, IFN-β and IFN-α2 (*Figure 8A–G*). Of note, despite the low levels of viral replication in HBE infected from the apical surface (*Figure 4*), we observed near-equivalent levels of IFN induction under these conditions (*Figure 8A–G*). At 48 hr p.i., levels of type III IFNs further increased to very high levels (>10 ng/ml)

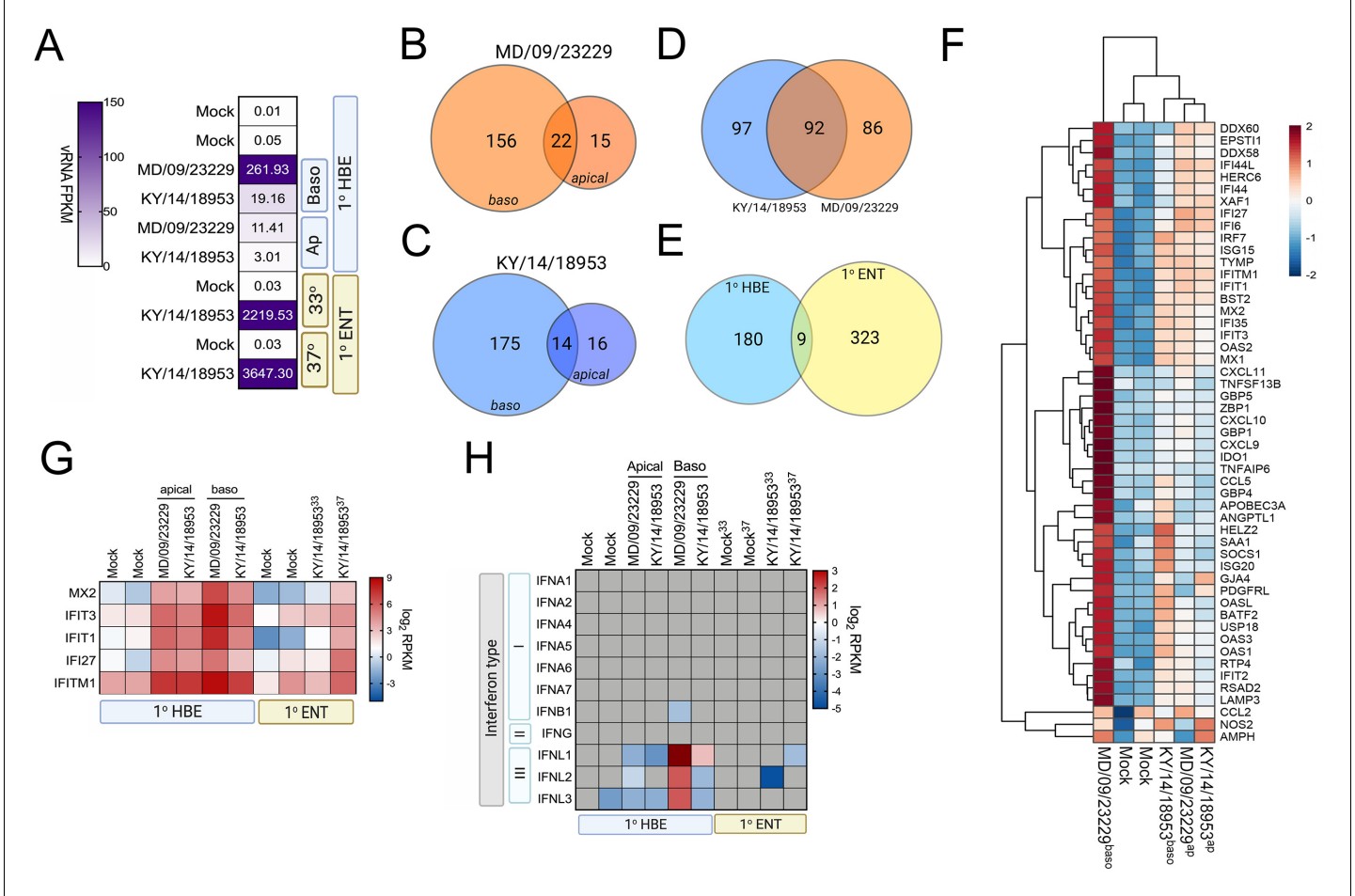

**Figure 7.** EV-D68 infection of primary human airway cells induces robust antiviral signaling. Whole-genome RNAseq-based transcriptional profiling from total RNA isolated from primary human bronchial epithelial (HBE) cells grown at an air–liquid interface or primary human enteroids infected with EV-D68 isolates MD/09/23229 and KY/14/18953 was performed in HBE infected from the apical or basolateral domains of in human enteroids infected at 33°C or 37°C. (A) Heatmap of vRNA FPKM (fragments per kilobase per million reads mapped) values for apical and basolateral infection of HBE with the indicated isolate and of enteroid infection at 33°C and 37°C with KY/14/18953. Key is at right. Purple indicates high viral reads, and white indicates low viral reads. (B–E) Venn diagrams denoting the overlap in differentially regulated transcripts in HBE infected from the apical or basolateral domains with MD/09/23229 (B) or KY/14/18953 (C) shared between both isolates following basolateral infections (D) and between HBE and enteroids infected with KY/14/18953 (E). (F) Heatmap of select interferon-stimulated genes (ISGs) in primary HBE infected with the indicated isolates of EV-D68 from the apical of basolateral domains or in mock-infected controls. Scale at right. Red indicates higher expression and blue indicates lower expression. (G) Heatmap of transcripts upregulated in infected HBE and enteroids by both strains based on log$_2$ RPKM values. Key is at right. Red indicates higher RPKM values, blue represents low RPKM values, and gray represents no reads. (H) Heatmap of transcripts (based on log$_2$ RPKM) associated with type I, II, or III interferons (IFNs) in HBE cells infected apically and basolaterally at 33°C with the indicated strains or in enteroids infected with KY/14/18953 at 33°C or 37°C. Scale at right, red indicates higher RPKM values, blue represents low RPKM values, and gray represents no reads.

(*Figure 8A,D–G*). In addition, at the later time point, we observed a significant induction of IFN-β, but not IFN-α2 (*Figure 8A–C*). In contrast to EV-D68-infected HBE, EV-D68 infection in enteroids did not induce detectable changes in any of the cytokines tested, including IFNs (*Figure 8A and D–G*). These data further support that the airway and intestinal epithelium induce cell-type-specific responses to EV-D68 infection.

Prior reports have suggested that in vitro respiratory virus replication differences at 33°C and 37°C may be related to increased IFN responses at higher temperatures (*Foxman et al., 2015*). To determine whether differential temperature-dependent IFN responses explained differences in EV-D68 replication in HBE at 33°C and 37°C, we again utilized Luminex-based multiplex assays against 37 pro-inflammatory cytokines, including type I and III IFNs. To do this, we compared infection of

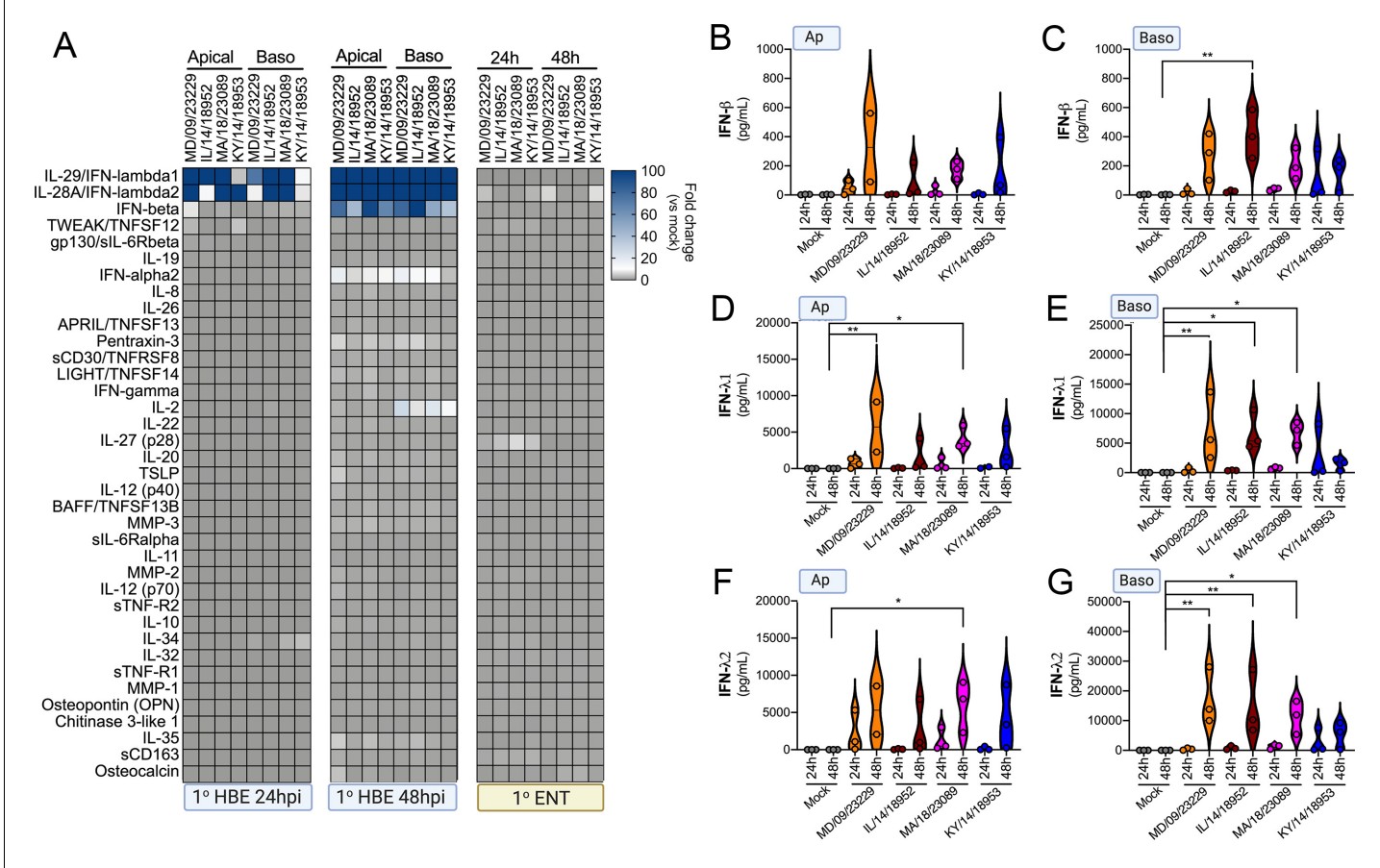

**Figure 8.** EV-D68 infection of primary human airway cells induces a preferential type III IFN response. (A) Luminex-based multianalyte profiling of 37 cytokines and chemokines in primary human bronchial (HBE) cells or enteroids (ENT) infected with $10^6$ PFU of the indicated EV-D68 strains MD/09/23229, IL/14/18952, MA/18/23089, or KY/14/18953 from the apical or basolateral surfaces and incubated at 33°C (HBE) or 37°C (ENT). Supernatants were collected from the apical compartment at 24 and 48 hr post-infection (hpi). Shown is a heatmap based on cytokines induced relative to mock-infected controls (key at right), with blue denoting significantly increased cytokines in comparison to uninfected. Gray denotes little to no change (scale at top right). Data are based on three independent experiments. Levels of IFN-β (B, C), IFN-λ1 (D, E), or IFN-λ2 (F, G) infected from the apical (B, D, F) or basolateral (C, E, G) are shown. Symbols represent individual biological replicates from unique donor cells. Statistical significance was determined using a Kruskal–Wallis test, *p<0.05, **p<0.01.

primary HBE cells with EV-D68 isolates MD/09/23229 and KY/14/18953 at 33°C and 37°C for either 24 or 48 hpi. Despite differences in the efficiency of replication in HBE at 33°C and 37°C, we did not detect any significant differences in the induction of type I (IFN-β) or III (IFN-λ1, IFN-λ2) IFNs under these conditions (*Figure 9A–D*). In addition, we found that Calu-3 lung epithelial and Caco-2 intestinal epithelial cell lines did not mount an IFN-mediated immune response to EV-D68 infection at either temperature (*Figure 9E,F*). These results suggest that temperature does not impact IFN signaling in response to EV-D68 infection of primary HBE.

## IFN signaling restricts EV-D68 replication in primary human airway cells

We observed robust IFN-mediated antiviral signaling in HBE cells infected with EV-D68 despite very low to undetectable levels of replication, suggesting that this antiviral response restricts EV-D68 infection. To test this, we infected HBE cells with EV-D68 in the presence of a selective small-molecule inhibitor of JAK1/2 signaling (ruxolitinib). Treatment of HBE with ruxolitinib significantly decreased the secretion of IFN-β and IFN-λ1 in response to EV-D68 infection (*Figure 10A,B*) and also significantly reduced ISG induction (*Figure 10C*). Consistent with this, there was a significant increase in MD/09/23229 infectious titers at 48 hpi as compared to dimethyl sulfoxide (DMSO)-treated controls, with less robust enhancement of KY/14/18953 (*Figure 10D*).

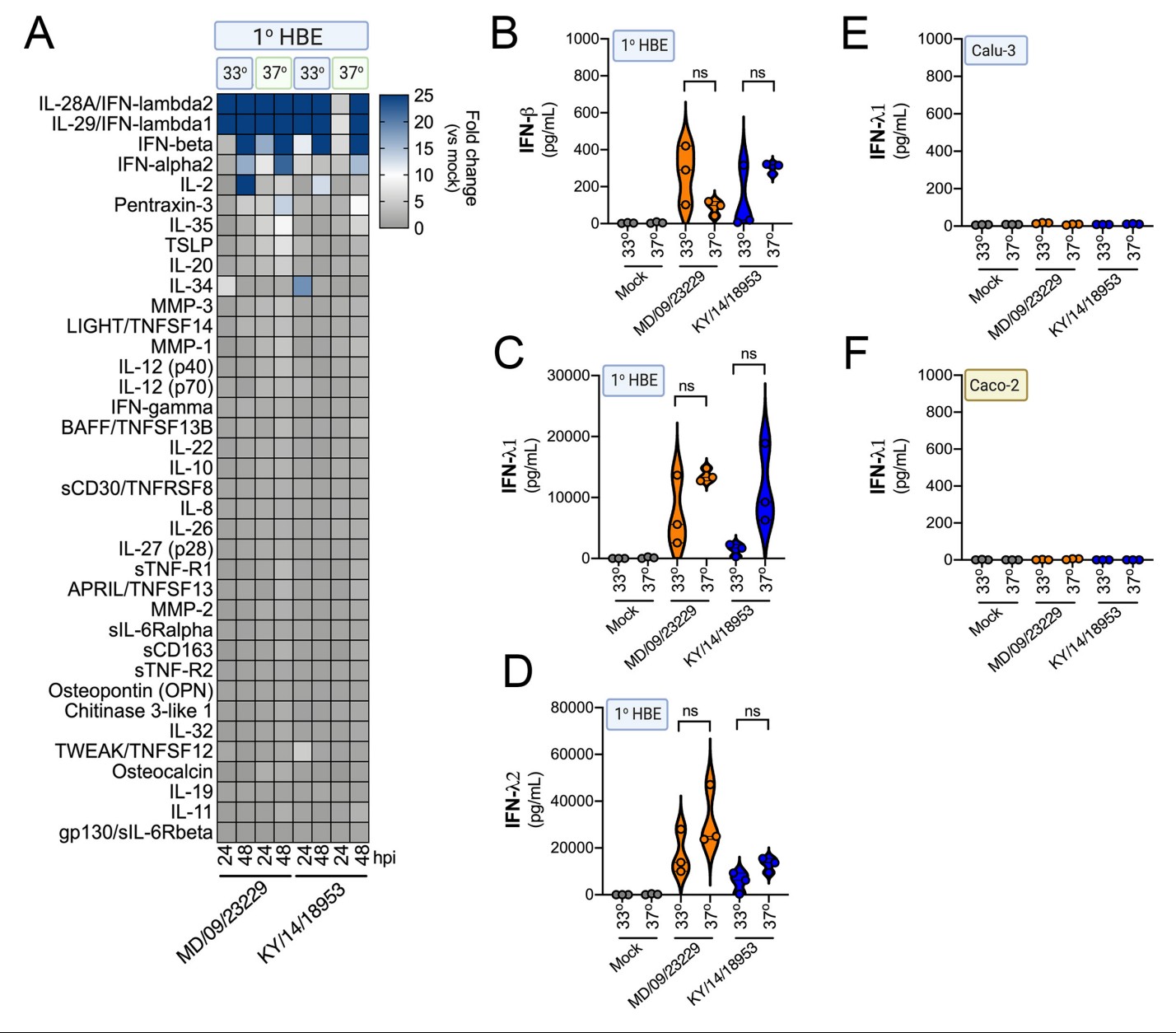

**Figure 9.** IFN induction in response to EV-D68 infection is independent of temperature. (A) Luminex-based multianalyte profiling of 37 cytokines and chemokines in primary human bronchial (HBE) cells infected with $10^6$ PFU of EV-D68 strains MD/09/23229 or KY/14/18953 at 33°C or 37°C. Supernatant was collected from the apical compartment at 24 and 48 hr post-infection (hpi). Shown is a heatmap based on cytokines induced relative to mock-infected controls (key at right), with blue denoting significantly increased cytokines in comparison to uninfected. Gray denotes little to no change (scale at top right). Data are based on three independent experiments. Levels of IFN-β (B), IFN-λ1 (C), or IFN-λ2 (D) from HBE infected at 33°C or 37°C are shown. Symbols represent individual biological replicates from unique donor cells. (E, F) Levels of IFN- λ1 as determined by Luminex-based assays in Calu-3 (E) or Caco-2 (F) cells infected with MD/09/23229 or KY/14/18953 at 33°C or 37°C. Symbols represent individual biological replicates. Statistical significance was determined using a Student's t-test, not significant (ns).

## Discussion

In this study, we defined differences in the dynamics of EV-D68 replication and pH stability using a panel of isolates from AFM peak years and a pre-outbreak isolate. In addition, utilizing two primary human cell models representing common tissue sites targeted by enteroviruses in humans, we

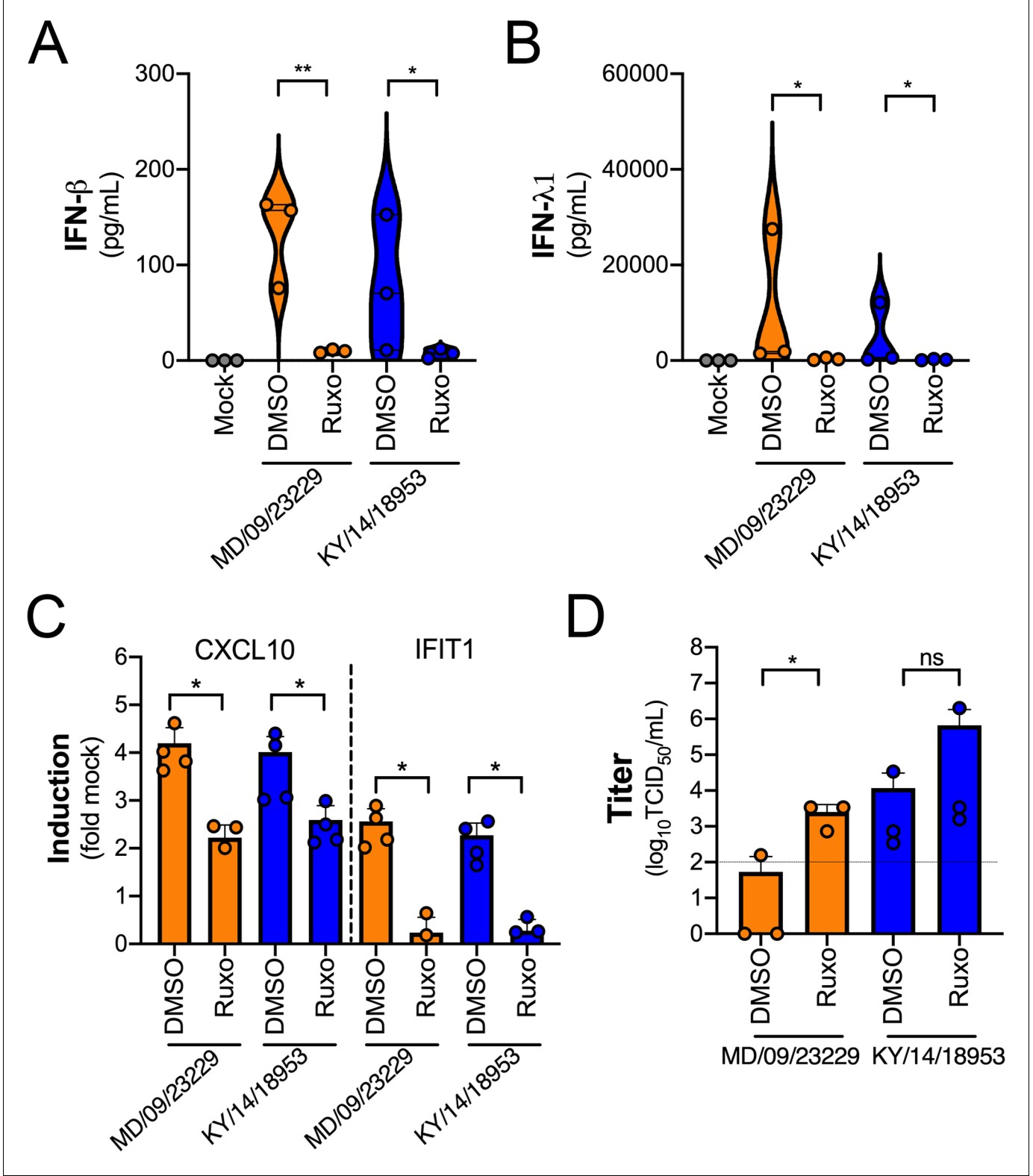

**Figure 10.** IFN signaling restricts EV-D68 replication in primary human airway cells. (**A, B**) Levels of IFN-β (**A**) or IFN-λ1 (**B**) as determined by Luminex-based assays in medium harvested from primary human bronchial epithelial (HBE) cells pretreated with the JAK1/2 inhibitor ruxolitinib (5 mM, Ruxo) or DMSO control for 1 hr and then infected with EV-D68 strains MD/09/23229 or KY/14/18953 as indicated in the presence of inhibitor for 48 hr at 33°C. Symbols represent individual biological replicates from at least two unique donors. (**C**) Induction of the interferon-stimulated genes (ISGs) CXCL10 or IFIT1 in control (DMSO)- or Ruxo-treated HBE infected with the EV-D68 strains MD/09/23229 or KY/14/18953 as assessed by RT-qPCR. Symbols

*Figure 10 continued on next page*

*Figure 10 continued*

represent individual biological replicates from at least two unique donors. (**D**) Viral titers in control (DMSO) or Ruxo-treated HBE infected with MD/09/23229 or KY/14/18953 for 48 hr at 33°C. Statistical significance was determined using a Student's t-test, *p<0.05, **p<0.01, ns, not significant.

defined differences in epithelial responses to EV-D68 between the respiratory and GI tracts. Collectively, this work details the varied responses of the respiratory and intestinal epithelium to historic and contemporary EV-D68 isolates and defines the role of type III IFN signaling in the control of EV-D68 infection in the respiratory, but not intestinal, epithelium.

We found that most isolates of EV-D68 efficiently replicated in both respiratory and intestinal epithelial cell lines, although there were some isolate-specific differences in temperature sensitivity. By comparison, primary HBE cells were less permissive to EV-D68 infection. One isolate, KY/14/18953, replicated very efficiently in primary HBE from either the apical or basolateral domains at both 33°C and 37°C, but exhibited a preferential release into the apical compartment. Another isolate, IL/14/18952, infected preferentially from the basolateral surface, but similarly was released into the apical compartment. The historic isolate MD/09/23229 and contemporary isolate MA/18/23089 replicated to comparably lower levels in all conditions but shared an apical release preference. Viruses with respiratory tropism as well as those with enteric tropism vary in their capacity to infect via the apical or basolateral surfaces, and while this information does not dictate which system is most susceptible to infection, it can inform future studies of cellular entry and receptor determination (*Bomsel and Alfsen, 2003*). As for viral egress, basolateral release is often associated with viral dissemination throughout the body, while apical release is associated with continued presence at the mucosal surface which can contribute to viral shedding and subsequent transmission to a new host (*Owusu et al., 2021*). Despite high levels of infection in the intestinal-derived Caco-2 cell line, only one isolate, KY/14/18953, replicated to high titers in stem cell-derived enteroids, although the contemporary isolate MA/18/23089 also replicated to low, but detectable, levels which improved when access to the apical surface was available. These studies point to key differences in the susceptibility of different primary epithelial-derived cell models to EV-D68 infection and suggest that host factors likely influence this tropism. For example, although attachment factors have been identified for EV-D68, including sialic acid and decay accelerating factor, some contemporary strains of EV-D68 including KY/14/18953 do not bind to sialylated receptors and their role in mediating infection is unknown (*Baggen et al., 2016*; *Blomqvist et al., 2002*). In addition, the neuron-specific intercellular adhesion molecule 5 (ICAM-5/telencephalin) has been identified as a potential receptor for several historic and contemporary isolates of EV-D68, but restricted expression in other cell types makes it unclear what role it might play in the epithelium (*Hixon et al., 2019*; *Wei et al., 2016*).

Previous work with EV-D68 before the emergence of AFM suggested that due to preferences for replication at 33°C and sensitivity to acid in vitro, it was more suited to be a respiratory pathogen behaving similarly to rhinoviruses as opposed to other enteroviruses (*Liu et al., 2018*; *Oberste et al., 2004*). While previous studies have evaluated acid stability of EV-D68, we utilized biologically relevant solutions with complexities other than acidity, such as bile acid and phospholipids, that more closely mimic the GI environment. Our studies using multiple contemporary isolates after the emergence of AFM suggest that these isolates are relatively stable at 37°C and also have improved acid stability. However, none of the EV-D68 isolates tested were stable during even short incubations with the most acidic fluid, the simulated fasted state stomach fluid, at a pH of 2. Our data also indicate that many isolates of EV-D68, even those associated with AFM outbreaks, are unable to replicate efficiently in human enteroids. However, one contemporary isolate, KY/14/18953, replicated to high levels in human enteroids and we observed replication of the contemporary isolate MA/18/23089 when primary intestinal enteroids were cultured on Transwell inserts. The basis for the very high capacity of KY/14/18953 to replicate in enteroids is unknown, but this isolate is genetically unique, and it is one of the very few members of the newly defined clade D, which thereby exhibits significant sequence variation in the VP1 region often used for receptor binding. These data suggest that the intestinal epithelium might serve as a site of EV-D68 transmission, particularly for some isolates.

The factors that mediate EV-D68 infection in the epithelium remain largely unknown, but our studies suggest that the induction of IFN signaling plays a major role in restricting replication in the airway epithelium. We have shown previously that type III IFNs are preferentially induced by

enterovirus infections in human enteroids and that this signaling restricts replication (*Drummond et al., 2017*; *Good et al., 2019*). In addition, type III IFNs are also the dominant IFNs induced in response to influenza, RSV, measles, and mumps infections of respiratory epithelial cells (*Crotta et al., 2013*; *Fox et al., 2015*; *Galani et al., 2017*; *Jewell et al., 2010*; *Okabayashi et al., 2011*). Although the type I IFN IFN-β was induced in response to EV-D68 infection, its induction was delayed compared to type III IFNs. Of note, we observed significant induction of IFNs even when levels of infection were not detectable, highlighting the potency by which the airway epithelium responds to these infections. The induction of IFNs is likely one mechanism by which the airway restricts EV-D68 replication, which is supported by our findings that treatment of HBE with ruxolitinib increased viral replication. However, it should be noted that ruxolitinib only partially recovered infection, suggesting that other cellular pathways in addition to IFNs also restrict infection. Surprisingly, despite robust IFN induction in response to EV-D68 infection of primary HBE, primary human enteroids did not mount any detectable IFN response to EV-D68 infection, suggesting that there are important differences in the capacity of the respiratory and airway epithelium to sense and respond to EV-D68 infection. The lack of antiviral signaling in infected enteroids would appear to be specific for EV-D68, as we have shown previously that enteroids infected with other enteroviruses including CVB, echoviruses, and EV71 respond via the induction of type III IFNs (*Drummond et al., 2017*; *Good et al., 2019*). While the mechanistic basis for this is unknown, differences in viral antagonism strategies and/or host detection mechanisms may explain these differences.

There are currently no available virus-specific treatments or vaccines to prevent AFM, which is an important emerging illness with significant morbidity to young children. Further understanding how EV-D68 targets the airway and/or GI epithelium are critical to improve our understanding of how this virus is transmitted, particularly given increases in its circulation. Our work presented here provides important insights into the dynamics of EV-D68 replication in the human airway and intestinal epithelium and provide ideal models to develop and test anti-EV-D68 therapeutics.

## Materials and methods

### Cell culture
HeLa 7B cells were provided by Dr. Jefferey Bergelson, Children's Hospital of Philadelphia, Philadelphia, PA, and grown in MEM, with 5% FBS, non-essential amino acids, and penicillin/streptomycin. Calu-3 cells (HTB-55, RRID:CVCL_0609) were obtained from the ATCC and grown in MEM w/ 10% FBS and 1% pen/strep and Caco-2 cells (BBE clone, CRL-2101) were obtained from the ATCC and grown in Dulbecco's MEM (DMEM) with 10% FBS and 1% pen/strep. All cell lines were tested for mycoplasma on a monthly basis using a sensitive PCR-based assay (Southern Biotech, 13100–01).

### Human intestinal enteroids
Human intestinal enteroid lines were derived as previously described by isolation of intestinal crypts from small intestines (*Drummond et al., 2017*) obtained from the University of Pittsburgh Biospecimen Core through an honest broker system after approval from the University of Pittsburgh Institutional Review Board and in accordance with the University of Pittsburgh anatomical tissue procurement guidelines and frozen. Enteroid lines were thawed, passaged, and maintained as previously described (*Stewart et al., 2020*) in Matrigel. Experiments with enteroids were performed on a Matrigel coating or on Transwell inserts, as detailed in the text. Crypt culture medium was composed of Advanced DMEM/F12 (Invitrogen) with 20% HyClone ES (embryonic stem) Cell Screened Fetal Bovine Serum (Thermo Fisher Scientific), 1% penicillin/streptomycin (Invitrogen), 1% L-glutamine, 1% *N*-acetylcysteine (100 mM; Sigma-Aldrich), 1% N-2 supplement (100×; Invitrogen), 2% B27 supplement (50×; Invitrogen), Gibco Hepes (*N*-2-hydroxyethylpiperazine-*N*-2-ethane sulfonic acid, 0.05 mM; Invitrogen), ROCK Inhibitor Y-27632 (1 mM, 100×; Sigma) and supplemented with the following growth factors: WNT3a, R-spondin, and Noggin as produced by preconditioned media from WRN cells obtained from the ATCC (CRL-3276, RRID:CVCL_DA06) and described previously (*Miyoshi and Stappenbeck, 2013*) and hEGF (50 ng/ml; Thermo Fisher Scientific) (*Egan et al., 2016*; *Shaffiey et al., 2016*) and was changed every 48–72 hr throughout culturing.

## HBE cells

Primary HBE cells were differentiated from human lung tissue by following an IRB-approved protocol and were maintained at an air–liquid interface with differentiation media changed twice per week, as described previously (*Myerburg et al., 2010*). Differentiation media (BEGM/Ultroser G; Pall Corporation, Crescent Chemical Company, Islandia, NY) was comprised of 5 µg/ml insulin, 10 µg/ml transferrin, 0.07 µg/ml hydrocortisone, 0.6 µg/ml epinephrine, 0.8% vol/vol bovine hypothalamus extract, 0.5 mg/ml BSA, 0.5 µM ethanolamine, 15 ng/ml retinoic acid, 0.5 ng/ml human epidermal growth factor, 10 nM triiodothyronine, 0.5 µM phosphoethanolamine, and 0.5% vol/vol Ultroser G (USG) in DMEM/F12. Cells were cultured for 3–6 weeks in order to differentiate and achieve a mucociliary phenotype on phase contrast microscopy prior to all experiments. Mucus was removed by extensive washes in $1\times$ PBS prior to infection.

## Viruses and infections

Experiments were performed with a panel of EV-D68 viruses described in *Supplementary file 1*. Viruses were grown in HeLa cells at 33°C in 5% $CO_2$ until CPE was observed, purified by ultracentrifugation over a 30% sucrose cushion as previously described (*Morosky et al., 2016*). Purity of all viral stocks was confirmed by Sanger sequencing of VP1 using enterovirus-specific primers, as described previously (*Oberste et al., 2003*). Plaque assays were performed in HeLa cells overlayed with 1% agarose, incubated for 72 hr, and plaques counted after staining with crystal violet. Viruses were obtained from the ATCC or were provided by the Center for Disease Control and Prevention (CDC) as detailed in *Supplementary file 1*.

For infections, cells were infected with $10^6$ plaque-forming units (PFU) of indicated viral isolates. Virus was adsorbed to the cell surface (apical or basolateral as indicated) for 1 hr at room temperature, cells were then washed with PBS, and then media replaced prior to placement back in the incubator at the indicated temperature for the indicated times. For viral replication analysis, aliquots of media were collected at indicated times post-infection and virus was detected via TCID50 assays in HeLa cells in technical triplicate. For HBE growth experiments, media was applied to the apical surface at the indicated timepoint and incubated for 30 min at the experimental temperature prior to collection.

## Viral sequence analysis

Sequences of each viral isolate were obtained from NCBI GenBank, with accession numbers listed in *Supplementary file 1*. The phylogenetic tree was constructed using Mega X (RRID:SCR_000667) (*Kumar et al., 2018*; *Stecher et al., 2020*). Evolutionary history was constructed with the Neighbor-Joining method (*Saitou and Nei, 1987*), with the optimal tree visualized. Evolutionary distances were computed via Jukes-Cantor (*Jukes and Cantor, 1969*). Protein alignment was conducted with Jalview version 2 (RRID:SCR_006459) (*Waterhouse et al., 2009*).

## Simulated intestinal fluids

Simulated gastric fluid powders fasted state gastric fluid (FaSSGF), fasted state small intestinal fluid (FaSSIF), and fed state small intestine (FeSSIF) (Biorelevant, FFF02) were prepared as described by the manufacturer. $10^6$ PFU/ml of the indicated virus was incubated in FaSSGF, FaSSIF, FeSSIF, or DMEM for the indicated time at 37°C. A 1 ml aliquot was collected and neutralized to pH 7.0 with 2.5 M sodium hydroxide, and then replication competence was assessed via TCID50 assay.

## Quantitative PCR

Total RNA was isolated from cells using the Sigma GenElute Total Mammalian RNA Miniprep Kit (Sigma, RTN350), according to the manufacturer protocol with the addition of a Sigma DNase digest reagent (Sigma, DNASE70). RNA (1 µg total) was reverse transcribed using iScript cDNA Synthesis Kit (Bio-Rad, 1708891) and diluted to 100 µl in ddH20 for subsequent qPCR analyses. RT-qPCR was performed using the iQ SYBR Green Supermix or iTaq Universal SYBR Green Supermix (Bio-Rad, 1725121) on a CFX96 Touch Real-Time PCR Detection System (Bio-Rad, 1855195). Gene expression was determined on the basis of a $\Delta C_Q$ method and normalized to human actin. Primer sequences can be found in *Supplementary file 3*.

## RNASeq

Total RNA was extracted as described above. RNA quality and concentration were determined by NanoDrop and then 1 μg of RNA was used for library preparation with TruSeq Stranded mRNA Library Preparation Kit (Illumina) per the manufacturer's instructions. Illumina NextSeq 500 was used for sequencing. Data were processed and mapped to the human reference genome (hg38) using CLC Genomics Workbench 20 (Qiagen, RRID:SCR_011853). Differential gene expression was analyzed with the DESeq2 package in R (RRID:SCR_015687) (*Love et al., 2014*). Raw sequencing files have been deposited in Sequence Read Archives and are publicly available (PRJNA688898).

## Luminex assays

Luminex profiles utilized the Human Inflammation Panel 1 37-plex assay kit (Bio-Rad, 171AL001M) per the manufacturer's protocol using the laboratory multianalyte profiling system (MAGPIX, Millipore) developed by Luminex Corporation (Austin, TX).

## Inhibitor treatments

HBE cells or enteroids on MG coats were incubated with 5 μM ruxolitinib (Sigma, ADV390218177) or DMSO control for 1 hr at 37°C and then infected with the indicated EV-D68 isolate in the presence of ruxolitinib or DMSO.

## Statistics

Statistical analysis was performed with GraphPad Prism software version 8.4.3 (RRID:SCR_002798). Experiments were performed at least three times, and primary cells from at least two genetically distinct donors were utilized for each experiment. Data are presented as mean ± standard deviation. Student's t-test or one-way analysis of variance (ANOVA) was used to determine significance, as indicated in figure legends, for normally distributed data. Growth curve analysis was completed using two-way ANOVAs. p-values of <0.05 was considered significant and are indicated in the figure legends.

## Acknowledgements

This work was supported by NIH R01-AI081759 (CBC), the Children's Hospital of Pittsburgh of the UPMC Health System (CBC), NIH T32-AI060525 (AIW), NIH F31-AI149866 (AIW), the Pediatric Infectious Diseases Society/St. Jude Research Hospital Fellowship in Basic and Translational Research (MCF), and the Cystic Fibrosis Foundation Research Development Program (University of Pittsburgh). All authors declare that they have no competing interests. All data needed to evaluate the conclusions in the paper are present in the paper and/or Supplementary Materials. Additional data related to this paper may be requested from the corresponding author. The findings and conclusions in this report are those of the author(s) and do not necessarily represent the official position of the Centers for Disease Control and Prevention or other contributing agencies.

## Additional information

### Funding

| Funder | Grant reference number | Author |
|---|---|---|
| National Institute of Allergy and Infectious Diseases | AI081759 | Carolyn B Coyne |
| National Institute of Allergy and Infectious Diseases | AI060525 | Alexandra I Wells |
| National Institute of Allergy and Infectious Diseases | AI149866 | Alexandra I Wells |
| Pediatric Infectious Diseases Society | Fellowship in Basic and Translational Research | Megan Culler Freeman |

The funders had no role in study design, data collection and interpretation, or the decision to submit the work for publication.

## Author contributions
Megan Culler Freeman, Conceptualization, Data curation, Formal analysis, Funding acquisition, Validation, Investigation, Visualization, Methodology, Writing - original draft, Writing - review and editing; Alexandra I Wells, Conceptualization, Data curation, Formal analysis, Investigation, Methodology, Writing - review and editing; Jessica Ciomperlik-Patton, Michael M Myerburg, Resources, Writing - review and editing; Liheng Yang, Resources, Methodology; Jennifer Konopka-Anstadt, Resources; Carolyn B Coyne, Conceptualization, Data curation, Formal analysis, Supervision, Funding acquisition, Validation, Investigation, Visualization, Methodology, Writing - original draft, Writing - review and editing

## Author ORCIDs
Alexandra I Wells (iD) https://orcid.org/0000-0001-8178-0376
Carolyn B Coyne (iD) https://orcid.org/0000-0002-1884-6309

## Decision letter and Author response
Decision letter https://doi.org/10.7554/eLife.66687.sa1
Author response https://doi.org/10.7554/eLife.66687.sa2

# Additional files

## Supplementary files
• Supplementary file 1. VP1 Sequence Alignment of EV-D68 isolates used in this study. Protein sequence alignment of the VP1 isolates used in the study, with MD/09–23229 used as the reference isolate (shown in top row). The consensus amino acid is denoted in the bottom row. Alignment constructed using Jalview with sequence information from GenBank.

• Supplementary file 2. Table of viral isolates used in the study.

• Supplementary file 3. Table of RT-qPCR primers used in this study.

• Supplementary file 4. Transcripts induced by apical or basolateral infection in HBE with MD/09/23229 (orange) and KY/14/18953 (blue).

• Supplementary file 5. Transcripts shared between basolateral infection of HBE with MD/09/23229 and KY/14/18953.

• Supplementary file 6. Transcripts induced by infection of human enteroids with KY/14/18953.

• Transparent reporting form

## Data availability
Raw sequencing files have been deposited in Sequence Read Archives and are publicly available (PRJNA688898).

The following dataset was generated:

| Author(s) | Year | Dataset title | Dataset URL | Database and Identifier |
|---|---|---|---|---|
| Freeman MC | 2021 | Respiratory and intestinal epithelial cells exhibit differential innate immune responses to contemporary EV-D68 isolates | https://www.ncbi.nlm.nih.gov/bioproject/PRJNA688898/ | NCBI BioProject, PRJNA688898 |

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
