## [Decision Letter]

**Acceptance summary:**

Is enterovirus D68, currently endemic in the US and many countries in Asia, an enteric or respiratory pathogen? Surprisingly, different isolates have remarkably different phenotypes in cell lines, polarized cells and organoid cultures, supporting the argument that EV68 infection can either be a respiratory or enteric pathogen, which might contribute to its somewhat confounding epidemiology. This is an important paper that supports the interesting concept that EV-D68 is currently evolving for both respiratory and fecal/oral spread.

**Decision letter after peer review:**

Thank you for submitting your article "Respiratory and intestinal epithelial cells exhibit differential susceptibility and innate immune responses to EV-D68" for consideration by *eLife*. Your article has been reviewed by 2 peer reviewers, one of whom is a member of our Board of Reviewing Editors, and the evaluation has been overseen by Sara Sawyer as the Senior Editor. The following individual involved in review of your submission has agreed to reveal their identity: Shin-Ru Shih (Reviewer #2).

Essential revisions:

The data that support the differential infectivities of different EV68 isolates are strong. Findings of the authors are discussed below, accompanied by requests for clarification and/or improvement of the presentation.

Figure 1: Six different viral isolates show differential temperature- and cell specificity in cultured lung and intestinal cells. Careful time courses are shown with infectivity measured with good statistics. There is a nice table in Figure 1C that summarizes this, but confusing because, as in the other tables orange, used in the figure for historical isolate MD/09/23229, is a different virus in the table. It would also be good to see panels in which the early time points are more spread out so that we could ascertain any differences between the first infectious cycle and subsequent spread.

Figure 2. If EV-D68 replicates in the intestine, it should be insensitive to conditions that mimic the small intestine or even the stomach. The six isolates are now winnowed to four based on the categories identified in Figure 1: one growing well under all conditions (KY), the historical isolate that grows primarily in lung cells (MD), primarily in intestinal cells (MA) and primarily at low temperatures (IL). Interestingly, all viruses except the MD strain are resistant to incubation in small intestine-mimicking solutions.

Figure 3 shows comparative growth in primary lung epithelial cells and compared apical vs basolateral infection and release. While the data are good, the reader is not helped with the interpretation of these interesting but confusing results – the cell model figures only illustrate entry, but the figure shows entry and release.

Figure 4 shows growth in primary enteroids from two different individuals. The KY strain, which grows under most previous conditions, does in organoids too. The other three strains don't grow in the organoids of one donor. That growth in the enteroids of the second donor are not tested for all samples weakens any arguments made about including these data.

Figure 5 shows RNA-SEQ data. Differential induction of antiviral responses are shown, but discussed in such a confusing way that the conclusions become unclear. This section needs to be re-written to discuss one variable at a time, for the non-specialist.

Figure 6 shows that, as suggested from the genomics in Figure 5, in which titers of both the MD and KY isolate are shown to be susceptible to the inhibitor of type 1 IFN signaling in bronchial epithelial cells. However, this does not distinguish whether the infections are controlled by interferons α/β of λ, a conclusion that must be inferred from the RNA-SEQ data, and therefore more carefully explained. Given the format of *eLife*, some of the requisite data to make these arguments could be rescued from the Supplemental Figures.

In short, this is an important paper that supports the interesting concept that EV-D68 is currently evolving for both respiratory and fecal/oral spread. However, the significance of the findings are not given satisfactory explication and interpretation. pH-resistance is clear even to the non-expert audience of *eLife*, but, in terms of apical vs. basolateral entry and release, what is expected for fecal-oral vs. respiratory spread? This reviewer had to make a table. With respect to the suppression of host responses, some comments on comparative sequence in relevant genes, correlation with viral yield, etc. are necessary to interpret the results.

The chosen isolates are difficult for the non-specialist to follow. An evolutionary perspective is needed, and should be addressed by the inclusion of:

1. A sequence and/or evolutionary tree with the tested isolates identified.

2. Alignments of any viral sequences available in Supplemental Materials, with comments on the protein-coding sequences that are most variable, especially those concerned with host responses. Under normal circumstances, I would suggest that the authors perform sequence analysis of EV-D68 strains they studied and list the similarities and differences of the sequences. If any particular sequences may involve in the different phenotypes, then perform a mutagenesis study to confirm the findings.

3. Figure, 6D, ruxolitnib, a jak1/2 inhibitor was used to prove that inhibition of ISG signaling would reduce viral replication. The conclusion should be more cautious because the specificity of this inhibitor is not high. Other inhibitors should be tested too.

---

## [Author Response]

Essential revisions:The data that support the differential infectivities of different EV68 isolates are strong. Findings of the authors are discussed below, accompanied by requests for clarification and/or improvement of the presentation.Figure 1: Six different viral isolates show differential temperature- and cell specificity in cultured lung and intestinal cells. Careful time courses are shown with infectivity measured with good statistics. There is a nice table in Figure 1C that summarizes this, but confusing because, as in the other tables orange, used in the figure for historical isolate MD/09/23229, is a different virus in the table. It would also be good to see panels in which the early time points are more spread out so that we could ascertain any differences between the first infectious cycle and subsequent spread.

The tables have been updated to reflect the same color scheme as the graphs. In addition, we have amended the x-axis, as requested, to better represent the early time points.

Figure 3 shows comparative growth in primary lung epithelial cells and compared apical vs basolateral infection and release. While the data are good, the reader is not helped with the interpretation of these interesting but confusing results – the cell model figures only illustrate entry, but the figure shows entry and release.

The figure and associated table/diagrams have been modified to assist with result interpretation. In addition, we have clarified the discussion of these results to assist with interpretation as well.

Figure 4 shows growth in primary enteroids from two different individuals. The KY strain, which grows under most previous conditions, does in organoids too. The other three strains don't grow in the organoids of one donor. That growth in the enteroids of the second donor are not tested for all samples weakens any arguments made about including these data.

There may be some confusion about the interpretation of this figure that we have addressed both in the figure and in the text. The graphs in A and B correspond to the experimental condition of enteroids grown in Matrigel. Experiments are inclusive of at least two independent donor-derived, genetically diverse enteroid lines. The graphs in C-F correspond to the experimental condition of enteroids grown in a single layer on Transwell inserts, again inclusive of at least two independent donor-derived, genetically diverse enteroid lines. Only two strains of virus were tested in these experiments, as they had previously performed best at 37^o^C in the intestinal cell line (Caco-2, Figure 1). Modifications were made to the figure to be consistent with the format/table/diagram changes in revised Figure 3.

Figure 5 shows RNA-SEQ data. Differential induction of antiviral responses are shown, but discussed in such a confusing way that the conclusions become unclear. This section needs to be re-written to discuss one variable at a time, for the non-specialist.

Figure 5 (revised Figure 7) are data generated from multiplexed Luminex assays, not RNASeq. To eliminate confusion regarding the order of figures and the discussion of results, we have reincorporated the RNAseq data that was previously in the supplement. The RNASeq data are now found in the revised Figure 6, with the Luminex data in revised Figure 7. In addition, discussion of both RNASeq and Luminex has been edited for clarity.

Figure 6 shows that, as suggested from the genomics in Figure 5, in which titers of both the MD and KY isolate are shown to be susceptible to the inhibitor of type 1 IFN signaling in bronchial epithelial cells. However, this does not distinguish whether the infections are controlled by interferons α/β of λ, a conclusion that must be inferred from the RNA-SEQ data, and therefore more carefully explained. Given the format of eLife, some of the requisite data to make these arguments could be rescued from the Supplemental Figures.

Ruxolitinib abrogates signaling downstream if both type I and III IFNs, thus we agree with the reviewer that conclusion cannot be drawn regarding the contribution of either pathway. To address this concern, we have reorganized figures and relocated data previously in the Supplemental Figures to the main text. In addition, we have modified the discussion to provide a more integrated analysis of these findings.

In short, this is an important paper that supports the interesting concept that EV-D68 is currently evolving for both respiratory and fecal/oral spread. However, the significance of the findings are not given satisfactory explication and interpretation. pH-resistance is clear even to the non-expert audience of eLife, but, in terms of apical vs. basolateral entry and release, what is expected for fecal-oral vs. respiratory spread? This reviewer had to make a table. With respect to the suppression of host responses, some comments on comparative sequence in relevant genes, correlation with viral yield, etc. are necessary to interpret the results.

We have supplemented the discussion around these issues to assist with interpretation for a more general audience.

The chosen isolates are difficult for the non-specialist to follow. An evolutionary perspective is needed, and should be addressed by the inclusion of:1. A sequence and/or evolutionary tree with the tested isolates identified.2. Alignments of any viral sequences available in Supplemental Materials, with comments on the protein-coding sequences that are most variable, especially those concerned with host responses. Under normal circumstances, I would suggest that the authors perform sequence analysis of EV-D68 strains they studied and list the similarities and differences of the sequences. If any particular sequences may involve in the different phenotypes, then perform a mutagenesis study to confirm the findings.

To address this concern, we now include a figure displaying the evolutionary relationship between isolates (Figure 1 in the revised manuscript), which includes a phylogenetic tree (Figure 1A) as well as an overview of the amino acid differences in the polyprotein which demonstrates that VP1 has the greatest number of mutations from isolate to isolate (Figure 1B). We have added a protein sequence alignment as Supplemental Figure 1 (new numbering) showing differences in the VP1 region. We have added additional commentary on regions of conservation.

3. Figure, 6D, ruxolitnib, a jak1/2 inhibitor was used to prove that inhibition of ISG signaling would reduce viral replication. The conclusion should be more cautious because the specificity of this inhibitor is not high. Other inhibitors should be tested too.

Ruxolitinib is an FDA-approved selective JAK 1/2 inhibitor. While we agree with the reviewer that the specificity of this inhibitor to distinguish between type I and III IFN signaling is non-selective, it should be noted that Ruxolitinib is used clinically to target JAK signaling given its potency and specificity. We are not aware of more specific inhibitors that could target the pathways downstream of IFNs. Given the relatively non-tractable nature of primary HAE cultures, genetic-based approaches (e.g., RNAi, CRISPR-mediated deletion) are limited. However, we agree that given that we are unable to differentially target the type I and III IFN pathways using this inhibitor, we have amended the text to better reflect this limitation.